# Inference Stage Optimization for Cross-scenario 3D Human Pose Estimation

**Jianfeng Zhang   Xuecheng Nie   Jiashi Feng**

Department of ECE, National University of Singapore

zhangjianfeng@u.nus.edu niexuecheng@u.nus.edu elefjia@nus.edu.sg

## Abstract

Existing 3D human pose estimation models suffer performance drop when applying to new scenarios with unseen poses due to their limited generalizability. In this work, we propose a novel framework, Inference Stage Optimization (ISO), for improving the generalizability of 3D pose models when source and target data come from different pose distributions. Our main insight is that the target data, even though not labeled, carry valuable priors about their underlying distribution. To exploit such information, the proposed ISO performs geometry-aware self-supervised learning (SSL) on each single target instance and updates the 3D pose model before making prediction. In this way, the model can mine distributional knowledge about the target scenario and quickly adapt to it with enhanced generalization performance. In addition, to handle sequential target data, we propose an online mode for implementing our ISO framework via streaming the SSL, which substantially enhances its effectiveness. We systematically analyze why and how our ISO framework works on diverse benchmarks under cross-scenario setup. Remarkably, it yields new state-of-the-art of 83.6% 3D PCK on MPI-INF-3DHP, improving upon the previous best result by 9.7%.

## 1 Introduction

3D human pose estimation aims to localize 3D human body joints in images or videos. As a fundamental task in computer vision, it is widely applied to human-robot interaction [15], action recognition [53], human tracking [32], etc. This task is commonly resolved in a fully-supervised manner with golden annotations [30, 57, 32, 55] that are collected in well-controlled laboratorial environments [21]. Despite their success in constrained scenarios, these methods are hardly generalized to new scenarios (e.g., in-the-wild scenes), due to severe differences in the underlying distributions (e.g., varying poses, camera viewpoints, body sizes and appearances).

Recent works address such a generalization challenge by leveraging either data augmentation strategies such as image composition [32] and synthesis [8, 49], or more complicated model learning strategies like introducing kinematics priors [57, 12], separating 2D and depth features [44, 31, 45, 17] or adopting adversarial learning [54, 14, 51]. However, they are still limited to the cases where training and test samples have similar poses and otherwise tend to suffer large performance drop, since their trained model is commonly biased to the training distribution and hardly generalizes well to an unseen one that is very different.

In this work, we propose a novel scheme named Inference Stage Optimization (ISO). Instead of focusing on improving model training, ISO improves and adapts the model at its inference stage before making predictions (Fig. 1). Our insight is that the target samples, although not labeled, carry valuable information about their distribution, which could be exploited to help adapt the model in the inference stage for correcting unfavorable training bias and improving generalization performance.

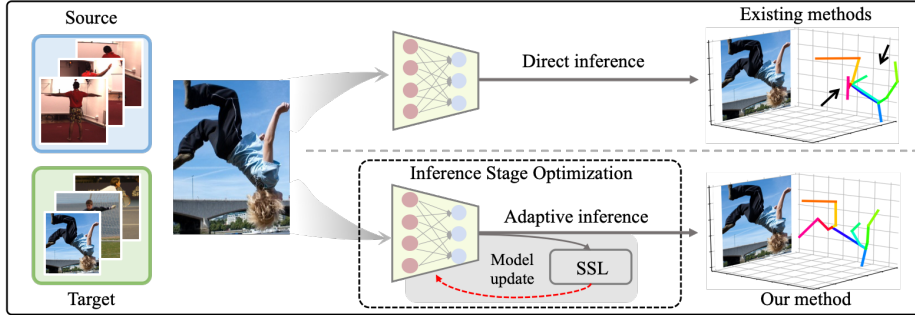

Figure 1: Illustration on our main idea. We consider cross-scenario setting where the model is trained on the source scenario (e.g., indoor scenes) but applied to a new target scenario (e.g., in-the-wild scenes). Existing methods (upper panel) usually use the trained model for predictions directly, which would suffer performance drop under such cross-scenario setup. Different from them, ISO adapts the model at its inference stage via performing self-supervised learning (SSL) on unlabeled target samples before making predictions (bottom panel), which largely improves its generalizability. Red arrow represents back-propagation based model update. Errors are labeled in black arrows.

However, exploiting such prior from unlabeled data is highly non-trivial. Inspired by recent success of self-supervised learning (SSL) techniques for learning good representations from unlabeled samples in other domains, we propose to leverage SSL to explore the underlying prior from unlabeled target instances. Different from general objects, human poses present clear and informative geometry structure, thus we deploy two different SSL methods, namely random projection adversary [14] and geometric cycle consistency [7], which are simple but effective at learning geometry-aware representations. ISO therefore enables the model to mine both geometric and distributional information from target instances and quickly adapt to the target scenario. As such, the model can estimate 3D poses more reliably across different scenarios, even in presence of severe distribution shifts.

Concretely, when training on labeled source data, instead of only performing fully-supervised learning (FSL) [30, 44, 55], our proposed ISO trains the model with FSL and SSL jointly. Such a training scheme enables the model to leverage geometry-wise feedback from SSL to learn representations and estimate 3D poses. This also facilitates model optimization in the inference stage. During inference, ISO adapts the model parameters to the new scenario and distribution via performing SSL on each target instance. Equipped with such instance-specific adaptation, the model can estimate 3D pose for each sample from the new target scenario accurately. In addition, we also develop an online ISO to accumulate the learned adaptation knowledge from a sequence of target samples, which would speed up model adaptation and reduce computational overhead.

We conduct extensive experiments under cross-scenario setup: training a model on Human3.6M [21] and evaluate it on MPI-INF-3DHP [31] and 3DPW [50]. Notably, ISO achieves new state-of-the-art accuracy, 83.6% 3D PCK on MPI-INF-3DHP, improving upon the previous best result by 9.7%.

Our contributions are four-fold. 1) To our best knowledge, we are among the first to explore the practical cross-scenario 3D pose estimation task and develop an effective solution (i.e., ISO). Distinguished from existing works, we explore how to effectively adapt the models during the inference stage. 2) We identify and investigate two simple SSL techniques suitable for 3D pose estimation under the ISO framework, to exploit geometric and distributional knowledge from unlabeled target data. 3) We develop an online ISO framework, which can handle sequential data effectively and naturally apply to practical scenes where data usually come online sequentially. 4) We provide understandings on why and how ISO works for cross-scenario generalization by conducting systematic analysis, which may inspire future works on improving generalization of human pose estimation.

## 2   Related work

**3D pose estimation.**   Lots of deep methods have been proposed for 3D pose estimation from 2D representations (e.g., images or poses) [48, 8, 49, 30, 44, 45, 16, 55, 34, 4, 40, 56], which highly rely on well-annotated datasets. These methods easily overfit to distribution-specific patterns such

as camera views and pose subjects, and can hardly generalize to new scenarios. To improve their generalizability, semi- and weakly-supervised methods [57, 54, 12, 51, 17, 52, 9, 36, 27] have been developed. Some [57, 12, 36] use kinematics priors for regularization or post-processing; others [54, 51] leverage adversarial training or separate 2D and depth features [44, 31, 45, 17] for domain adaptation. Despite encouraging results, the applicability of these methods is still restricted in scope defined by the datasets they are trained on. Recently, several geometry-driven self-supervised methods [39, 14, 7, 26, 37, 28, 38] are proposed to train the model with more unlabeled *training data*. However, they are rarely used in a transductive manner for testing. Different from all above methods, we are the first to learn distributional information from target instances at inference stage via SSL, which is demonstrated an effective method for out-of-distribution 3D pose estimation.

**Learning on target instances.** Learning on target instances has emerged as a powerful technique for mining complex data distributions and priors. Bau *et al.* [3] improve photo manipulation performance by adapting image priors to the statistics of an individual target image. Sun *et al.* [46] leverage rotation prediction pretext task for solving domain shift in image classification. Shocher *et al.* [41] perform super-resolution of a target image via learning to recover it from its downsampled counterpart. However, these methods cannot be directly applied to 3D pose estimation. In this work, we propose a novel ISO framework to improve 3D pose estimation under cross-scenario setup through mining geometric and distributional knowledge from target instances.

## 3 Method

### 3.1 Problem formulation

Let $\mathbf{I}$ denotes an image and $\boldsymbol{x} \in \mathbb{R}^{J \times 2}$ denotes 2D spatial coordinates of $J$ keypoints of the human in the image. $\boldsymbol{X} \in \mathbb{R}^{J \times 3}$ denotes the corresponding 3D joints position. We consider such cross-scenario setup: the model is trained on a source scenario $\mathcal{D}_s$ (e.g., indoor scenes) with pose distribution $\mathcal{P}$, and applied to a new scenario $\mathcal{D}_t$ (e.g., in-the-wild scenes) with unseen poses, viewpoints, body sizes and appearances drawn from a different distribution $\mathcal{Q}$.

Empirically, a pose distribution $\mathcal{P}$ can be disentangled to appearance and geometry factors [39]. The cross-scenario setup is faced with the pose distribution drift w.r.t. both of them. However, drift of appearance distribution can be well solved by powerful off-the-shelf 2D pose estimators. Thus we focus on addressing the drift w.r.t. pose geometry (i.e., poses, viewpoints, etc). We directly work with skeleton data and aim to obtain a 3D pose model that can lift 2D poses to 3D ones with good adaptive capability to a new scenario.

Suppose we have a pair of 2D and corresponding 3D poses $\{(\boldsymbol{x}_i, \boldsymbol{X}_i)\}_{i=1}^N$ drawn i.i.d. from the source distribution $\mathcal{P}$. Existing methods usually train a 3D pose model on these training samples and apply it directly on target samples drawn from the target distribution $\mathcal{Q}$. In particular, the model with parameter $\boldsymbol{\theta}$ is trained in a fully supervised learning (FSL) scheme:

$$\min_{\boldsymbol{\theta}} \frac{1}{N} \sum_{i=1}^N \mathcal{L}_f(\boldsymbol{x}_i, \boldsymbol{X}_i; \boldsymbol{\theta}), \tag{1}$$

where $\mathcal{L}_f$ is a fully-supervised loss. Generally, $\mathcal{L}_f$ is defined as mean squared errors (MSE) of the predicted and ground truth (GT) poses [30]. Several earlier works complement such a loss with a bone supervision loss [44, 55]. Accordingly, $\mathcal{L}_f$ is formulated as

$$\mathcal{L}_f = \|\boldsymbol{X} - \widetilde{\boldsymbol{X}}\|_2^2 + \|\boldsymbol{B} - \widetilde{\boldsymbol{B}}\|_2^2. \tag{2}$$

Here $\boldsymbol{X}$ and $\widetilde{\boldsymbol{X}}$ denote the GT and predicted 3D poses, respectively; $\boldsymbol{B}$ and $\widetilde{\boldsymbol{B}}$ denote the GT and predicted bone vectors computed from $\boldsymbol{X}$ and $\widetilde{\boldsymbol{X}}$, respectively [44]. The obtained model is typically biased to the training samples and thus suffers limited generalizability.

### 3.2 Inference stage optimization

We introduce our Inference Stage Optimization (ISO) framework that allows a 3D pose model to mine geometric and distributional knowledge from target instances during the inference stage, and adapt to new scenarios with improved generalization performance. For simplicity, we consider a 3D

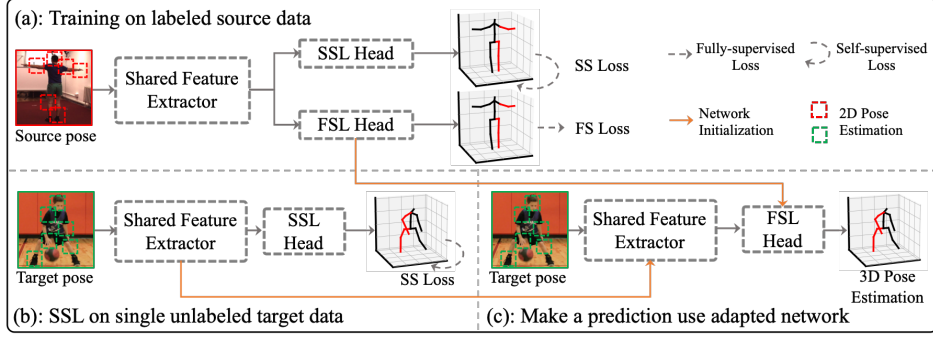

Figure 2: Overall pipeline of ISO. (a) We first train our model by solving optimization of both FSL and SSL tasks in the source scenario with labeled data. During inference, given each unlabeled target sample, (b) we first perform SSL on it to update network parameters and (c) exploit the adapted network for final pose estimation.

pose model implemented by a $K$-layer neural network with parameters $\theta_k$ for layer $k$. The stacked parameter vector $\boldsymbol{\theta} = (\theta_1, \ldots, \theta_K)$ specifies the entire model for 3D pose estimation. The overall pipeline is illustrated in Fig. 2.

### 3.2.1 Training

Similar to existing methods, when training on the source scenario $\mathcal{D}_s$, our model parameters $\boldsymbol{\theta}$ can be updated by solving the optimization problem in Eqn. (1). We call this the fully-supervised learning (FSL) task. However, our ISO also performs a self-supervised learning (SSL) task with self-supervised loss $\mathcal{L}_s(\boldsymbol{x})$ to train the pose estimation model so that it can learn to adapt via SSL feedback in the inference stage.

We choose two geometry-aware SSL methods to exploit pose geometry information from the skeleton data: random projection adversary [14] and geometric cycle consistency [7], which are effective at geometry adaptation. Note with our framework, more SSL methods can be explored in the future.

**ISO-Adversary.** The idea of random projection adversary SSL is that if a 2D pose is lifted to 3D accurately, and rotated and projected with randomly generated camera view, the resulting 'synthetic' 2D pose should lie within the valid 2D poses distribution. We build a pose discriminator $D$ to classify each input 2D pose as real or fake (randomly projected from 3D poses). The loss is defined as

$$\mathcal{L}_{adv} = \mathbb{E}(\log(D(\boldsymbol{r}))) + \mathbb{E}(\log(1 - D(\boldsymbol{y}))), \tag{3}$$

where $\boldsymbol{r}$ and $\boldsymbol{y}$ denote real and fake 2D poses, respectively. We follow [14] to generate random camera view by sampling an azimuth angle between $[-\pi, \pi]$ and an elevation angle between $[-\pi/9, \pi/9]$.

**ISO-Cycle.** The geometric cycle consistency SSL complements **ISO-Adversary** with cycle consistency among 2D and 3D spaces. Specifically, by lifting the randomly projected 2D pose $\boldsymbol{y}$ back to 3D and then re-projecting it to the original camera view, the resulting 3D and 2D poses should be consistent with the original ones. The training can thus be supervised by exploiting the cycle-consistency of the lift-project-lift process. Combined with the adversarial loss in Eqn. (3), the loss is

$$\mathcal{L}_{cyc} = \mathcal{L}_{adv} + \lambda_{2D}\|\boldsymbol{x} - \widetilde{\boldsymbol{x}}\|_2^2 + \lambda_{3D}\|\boldsymbol{X} - \widetilde{\boldsymbol{X}}\|_2^2, \tag{4}$$

where $\boldsymbol{x}$ and $\widetilde{\boldsymbol{x}}$ denote original and re-projected 2D poses, $\boldsymbol{X}$ and $\widetilde{\boldsymbol{X}}$ denote lifted and re-lifted 3D poses, $\lambda_{2D} = 10$ and $\lambda_{3D} = 0.1$ are weights for 2D and 3D loss terms, respectively.

During training, we optimize both FSL and SSL tasks to update network parameters. Following standard multi-task learning framework [5], the SSL task shares some of the network parameters $\boldsymbol{\theta}_e = (\theta_1, \ldots, \theta_\kappa)$ with the FSL task, where $\kappa \in \{1, \ldots, K\}$. We call these shared $\kappa$ layers as shared feature extractor. The SSL task uses its task-specific parameters $\boldsymbol{\theta}_s = (\theta'_{\kappa+1}, \ldots, \theta'_K)$. We call these unshared parameters $\boldsymbol{\theta}_s$ the SSL head, and $\boldsymbol{\theta}_f = (\theta_{\kappa+1}, \ldots, \theta_K)$ the FSL head. As shown in Fig. 2 (a), the joint architecture has a shared bottom and two heads. Both heads output a $J \times 3$ vector,

indicating the 3D pose prediction. The only difference between them is that their network parameters are updated by solving different optimization problems.

We train the model in a multi-task learning fashion on the same data drawn from $\mathcal{P}$. The joint-training problem is formulated as

$$\min_{\boldsymbol{\theta}_e, \boldsymbol{\theta}_f, \boldsymbol{\theta}_s} \max_{\boldsymbol{\theta}_d} \frac{1}{N} \sum_{i=1}^{N} \Big( \mathcal{L}_f(\boldsymbol{x}_i, \boldsymbol{X}_i; \boldsymbol{\theta}_e, \boldsymbol{\theta}_f) + \lambda \mathcal{L}_s(\boldsymbol{x}_i; \boldsymbol{\theta}_e, \boldsymbol{\theta}_s, \boldsymbol{\theta}_d) \Big). \tag{5}$$

where $\boldsymbol{\theta}_d$ denotes network parameters of the pose discriminator $D$ and $\lambda = 0.1$ is a relative weight for balancing different loss terms. Here $\mathcal{L}_s$ denotes the self-supervised loss in Eqn. (3) or Eqn. (4).

### 3.2.2 Inference

After minimizing Eqn. (5) on data from $\mathcal{D}_s$ with distribution $\mathcal{P}$, we obtain the network parameters $\boldsymbol{\theta}_e^{\mathcal{P}}, \boldsymbol{\theta}_f^{\mathcal{P}}, \boldsymbol{\theta}_s^{\mathcal{P}}$ and $\boldsymbol{\theta}_d^{\mathcal{P}}$ for the shared featured extractor, FSL head, SSL head and pose discriminator, respectively. During inference, ISO performs SSL on each single target instance $\boldsymbol{x}$ to update the shared feature extractor, SSL head and pose discriminator (Fig. 2 (b)), which can be formulated as

$$\min_{\boldsymbol{\theta}_e, \boldsymbol{\theta}_s} \max_{\boldsymbol{\theta}_d} \mathcal{L}_s(\boldsymbol{x}; \boldsymbol{\theta}_e, \boldsymbol{\theta}_s, \boldsymbol{\theta}_d). \tag{6}$$

The SSL process is done using standard gradient descent (or a variant) with learning rate $\alpha$ and iteration $T$. Additionally, a mini-batch contains several copies of $\boldsymbol{x}$ such that a single optimization iteration can involve adversarial samples (i.e., randomly projected 2D poses) as much as possible, which ensures better performance. After optimizing Eqn. (6), we obtain the updated parameter $\boldsymbol{\theta}_e^*$ of the shared feature extractor, and make a prediction using $\boldsymbol{\theta}^* = (\boldsymbol{\theta}_e^*, \boldsymbol{\theta}_f^{\mathcal{P}})$ (Fig. 2 (c)). The motivation behind this formulation is that the joint training scheme (FSL+SSL at the training phase) enables the FSL head to be adaptive to the representations learned from SSL. In this way, the FSL head, though being frozen, can be directly applied for making accurate predictions over the representations updated by the SSL branch during inference.

We implement ISO in a vanilla mode, i.e., performing SSL on each target instance individually before making prediction on it. For vanilla ISO, the optimization problem in Eqn. (6) is always initialized with parameters $\boldsymbol{\theta}_e^{\mathcal{P}}, \boldsymbol{\theta}_s^{\mathcal{P}}$ and $\boldsymbol{\theta}_d^{\mathcal{P}}$. After performing $T$ iterations SSL on $i^{th}$ instance $\boldsymbol{x}_i$, we obtain the updated parameters $\boldsymbol{\theta}_e^i, \boldsymbol{\theta}_s^i, \boldsymbol{\theta}_d^i$. After making a prediction on $\boldsymbol{x}_i$, $\boldsymbol{\theta}_e^i, \boldsymbol{\theta}_s^i$ and $\boldsymbol{\theta}_d^i$ are discarded.

Besides vanilla ISO, when the target instances arrive sequentially, we propose a corresponding online ISO by streaming the SSL to continuously exploit distributional knowledge among them. Specifically, the online ISO solves the same optimization problem to update network parameters. However, when learning on $\boldsymbol{x}_i$, $\boldsymbol{\theta}_e$, $\boldsymbol{\theta}_s$ and $\boldsymbol{\theta}_d$ are instead initialized with $\boldsymbol{\theta}_e^{i-1}, \boldsymbol{\theta}_s^{i-1}$ and $\boldsymbol{\theta}_d^{i-1}$ updated on the previous instance $\boldsymbol{x}_{i-1}$. This allows the model to benefit from the distributional information available in instances $\boldsymbol{x}_1, \ldots, \boldsymbol{x}_{i-1}$ as well as $\boldsymbol{x}_i$, and thus speeds up the model adaptation.

### 3.2.3 Network and algorithm details.

**Network architecture.** Our 3D pose estimation model primarily consists of the residual block (RB) proposed in [30]. Each RB consists of two linear layers, Batch Normalization (BN) [20], leaky ReLU [18] and dropout [42] with residual connection [19]. The feature dimension and dropout probability are set to 1,024 and 0.5, respectively. Specifically, the shared feature extractor consists of a linear layer followed by three stacked RBs. It first transforms the input $2J$-dimension vector to a 1024-dimension vector, which is then fed to the FSL and SSL heads separately. Both the FSL and SSL heads contain an unshared RB followed by a linear layer for 3D pose estimation. The pose discriminator takes as input the $2J$-dimension vector (2D pose) and outputs classification results (real or fake). We use three stacked RBs but remove all BN layers. For the 2-way classifier used for representation learning analysis (Sec. 4.3), we use the same architecture as the pose discriminator, except for the first layer since it takes 3D poses as inputs. The hidden feature used for visualization is extracted from the final residual block of the classifier (1024-dimension vector).

**Algorithm details.** The summary of both vanilla and online ISO on target instances during inference is illustrated in Algorithm 1.

**Algorithm 1:** Inference Stage Optimization.

---

**Input** : target instances $\{\boldsymbol{x}_i\}_{i=1}^N$, pre-trained network parameters $\boldsymbol{\theta}_e^{\mathcal{P}}, \boldsymbol{\theta}_f^{\mathcal{P}}, \boldsymbol{\theta}_s^{\mathcal{P}}, \boldsymbol{\theta}_d^{\mathcal{P}}$, learning rate $\alpha$,
&emsp;&emsp;&emsp;&emsp;training iteration $T$.
**Output** : 3D pose estimations $\{\boldsymbol{X}_i\}_{i=1}^N$.
**Initialization:** $\boldsymbol{\theta}_*^0 \leftarrow \boldsymbol{\theta}_*^{\mathcal{P}}$ with $* \in \{e, s, d\}$
**for** $i = 1$ *to* $N$ **do**
&emsp;**if** vanilla ISO **then**
&emsp;&emsp;$\boldsymbol{\theta}_*^i \leftarrow \boldsymbol{\theta}_*^0$ with $* \in \{e, s, d\}$
&emsp;**else**
&emsp;&emsp;// online ISO
&emsp;&emsp;$\boldsymbol{\theta}_*^i \leftarrow \boldsymbol{\theta}_*^{i-1}$ with $* \in \{e, s, d\}$
&emsp;**end**
&emsp;**for** $t = 1$ *to* $T$ **do**
&emsp;&emsp;Compute gradients $\nabla_{\boldsymbol{\theta}_*} \mathcal{L}_s(\boldsymbol{x}_i; \boldsymbol{\theta}_e^i, \boldsymbol{\theta}_s^i, \boldsymbol{\theta}_d^i)$ (Eqn. (6)) where $* \in \{e, s, d\}$.
&emsp;&emsp;Update parameters: $\boldsymbol{\theta}_*^i \leftarrow \boldsymbol{\theta}_*^i - \alpha \nabla_{\boldsymbol{\theta}_*} \mathcal{L}_s(\boldsymbol{x}_i; \boldsymbol{\theta}_e^i, \boldsymbol{\theta}_s^i, \boldsymbol{\theta}_d^i)$ where $* \in \{e, s, d\}$.
&emsp;**end**
&emsp;Predict 3D pose $\boldsymbol{X}_i$ using the network parameters $\boldsymbol{\theta}^i = (\boldsymbol{\theta}_e^i, \boldsymbol{\theta}_f^{\mathcal{P}})$ .
**end**

---

## 4 Experiments

We aim to answer the following questions through experiments: 1) Is ISO able to improve cross-scenario generalization performance of 3D pose estimation? 2) How does ISO take effect to boost generalization performance? 3) Does ISO introduce too much overhead in the inference stage?

### 4.1 Experiment setup

We quantitatively evaluate the generalizability of our method in cross-scenario setup, i.e., training a model on Human3.6M and evaluate its performance on the more challenging 3D pose benchmarks MPI-INF-3DHP [31] and 3DPW [50], which feature more diverse motions and scenes. We train our model on subjects S1, S5, S6, S7 and S8 of Human3.6M [30, 57] and evaluate it on the official test set of MPI-INF-3DHP and 3DPW. For MPI-INF-3DHP, we use Mean Per Joint Position Error (MPJPE), 3D Percentage of Correct Keypoints (PCK) with a threshold 150mm and the corresponding Area Under Curve (AUC) as metrics and adopt three evaluation protocols [17]: (i) *unscaled* (**US**); (ii) *glob. scaled* (**GS**); (iii) *procrustes* (**PA**). For 3DPW, we follow [24] to use Procrustes Aligned MPJPE (PA-MPJPE) and 3D PCK as metrics. In addition, we use MPII [1] and LSP [22], the standard 2D pose benchmarks with diverse scenes that reflect challenging factors such as strong pose deformations and abundant viewpoints in the real world, to qualitatively verify the effectiveness of our method.

We train our model for 200 epochs on Human3.6M, adopting Adam [25] as optimizer with an initial learning rate of $2 \times 10^{-4}$ and using exponential decay and mini-batch size of 1024. We use horizontal flip augmentation at both training and inference. During inference, for both 3DHP and 3DPW, we freeze batch normalization layers and perform SSL on each single target instance before making prediction. Specifically, for both vanilla and online ISO, we adopt Adam optimizer with learning rate $\alpha = 2 \times 10^{-5}$. We set iteration $T$ as 10 and 1 for vanilla and online ISO, respectively. In following experiments, unless otherwise stated we use **ISO-Cycle** SSL technique.

### 4.2 Does ISO boost generalization?

We compare ISO (**online**) with several state-of-the-art approaches on 3DHP and 3DPW datasets. Some methods consider domain adaptation [54], or use complex network architectures [30, 23, 12, 11, 55, 6, 13] and training schemes [51, 2, 47]. We use *Baseline* to denote the plain model trained with only FSL task; *Joint* is the model trained with FSL and SSL tasks jointly; *Vanilla* refers to the model adapted using vanilla ISO; *Online* is the model adapted using online ISO.

**Results on 3DHP.** We compare ISO against the methods in [54, 11, 6] under cross-scenario setup. We directly report their results from original papers. Note some of them have missing metrics or do not specify evaluation protocols. Additionally, we implement and compare with methods [55, 51]

based on their released code.[1] Table 1 shows the results under different metrics and protocols. Our method achieves the highest accuracy in terms of 3D PCK and MPJPE across all evaluation protocols, outperforming the second best by a large margin. This verifies the generalizability of our approach.

**Results on 3DPW.** We also compare ISO with state-of-the-art approaches on 3DPW. Some methods exploit temporal information [12, 24, 13], while some others are trained on the training set of 3DPW [2, 47]. Table 2 reports the results. Our method outperforms several approaches in terms of PA-MPJPE and even achieves comparable results with the fully-supervised method [47]. This shows the generalization capability of our method.

Table 1: Results on 3DHP. * denotes our implementation. US, GS and PA denote different protocols.

| Method | PCK ↑ | AUC ↑ | MPJPE ↓ |
|---|---|---|---|
| Yang [54] | 69.0 | 32.0 | - |
| Ci [11] | 74.0 | 36.7 | - |
| Chang [6] | 76.5 | 40.2 | - |
| Wandt [51] | 81.8 | 54.8 | 92.5 |
| Zhao [55]*(US) | 76.2 | 42.8 | 126.1 |
| Ours (US) | 83.6 | 48.2 | 92.2 |
| Zhao [55]*(GS) | 77.1 | 45.5 | 108.0 |
| Ours (GS) | 84.5 | 50.9 | 88.4 |
| Wandt [51]*(PA) | 81.6 | 47.0 | 95.4 |
| Zhao [55]*(PA) | 86.0 | 46.7 | 96.8 |
| Ours (PA) | 91.3 | 54.0 | 75.8 |

Table 2: Our results (14-joints) on 3DPW. * denotes training using GT data.

| Method | PCK ↑ | PA-MPJPE ↓ |
|---|---|---|
| Martinez [30] | - | 157.0 |
| Dabral [12] | - | 92.3 |
| Kanazawa [23] | 84.1 | 76.7 |
| Kanazawa [24] | 86.4 | 80.1 |
| Arnab [2]* | - | 77.2 |
| Doersch [13] | - | 74.7 |
| Sun [47]* | - | 69.5 |
| Ours | 82.0 | 70.8 |

**Qualitative results.** We visualize some 3D pose estimations of ISO on the challenging LSP, MPII, 3DHP and 3DPW datasets in Fig. 3. Most of the involved poses and camera views are unseen to our model. However, our ISO can still achieve good results even in presence of self-occlusion (1st column), large pose variations (2nd, 3rd column), and unusual views (4th column). Additionally, ISO compared with *Baseline* produces more geometrically plausible results. These verify the superior generalizability of ISO to challenging new scenarios.

**Components analysis.** Please refer to supplement for analysis of SSL techniques and hyper-parameters used in ISO.

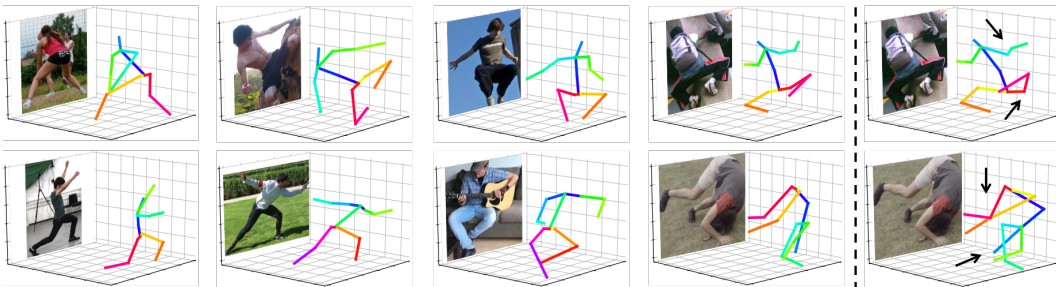

Figure 3: Example 3D pose estimations from LSP, MPII (top row) and 3DHP, 3DPW (bottom row). ISO results are shown in the left four columns. The rightmost column shows results of *Baseline*. Errors are labeled in black arrows. Please refer to supplement for more qualitative results.

## 4.3 Why ISO performs well?

We investigate why and how ISO can improve cross-scenario generalization. All below experiments are conducted on 3DHP using *Online* under the *unscaled* protocol, unless otherwise specified.

**Geometric distribution alignment.** Our main insight is performing ISO on target instances enables the model to mine geometric knowledge (e.g., limb length ratios and body parts symmetry) about the target distribution. To verify this, we inspect the distribution alignment in geometry of output

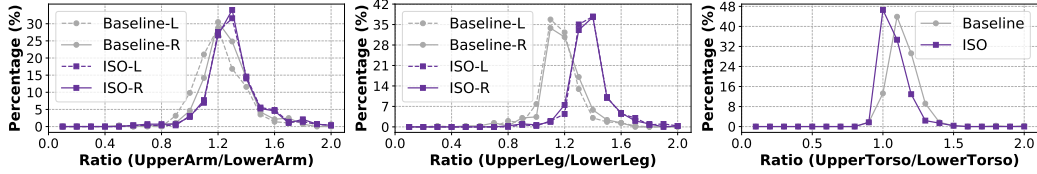

Figure 4: Distribution of limbs length ratio produced by ISO and *Baseline* on 3DHP. Left: Ratio of upper to lower arm. Middle: Ratio of upper to lower leg. Right: Ratio of upper to lower torso. Ground truth ratios are $\sim 1.3$, $\sim 1.3$ and $\sim 1.0$ for arm, leg and torso, respectively. **L** and **R** indicate left and right body parts, respectively.

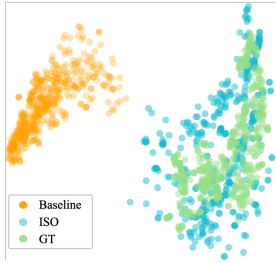
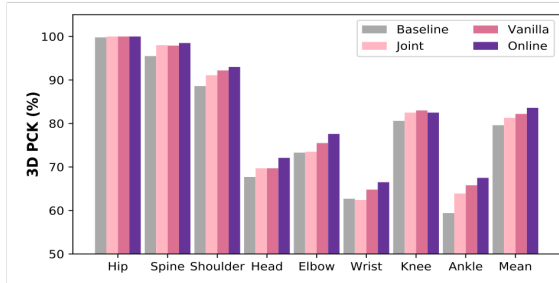

Figure 5: Visualization of hidden features using t-SNE.

Figure 6: Per body-part accuracy on 3DHP. PCK of each part is computed as the PCK of corresponding skeleton joints.

poses from *Baseline* and ISO (*Online*). Specifically, we compute the limb length ratios of upper to lower arms and legs (both for left and right sides), and torso [57, 7]. The results are shown in Fig. 4. We can observe the ratio distributions generated by ISO are sharper and closer to the real ratio distributions of 3DHP, compared with those by *Baseline*. Additionally, ISO produces more symmetric ratio distributions for the left and right sides of arms and legs than *Baseline*, which verifies its ability to capture the symmetry of body parts. All these results clearly demonstrate the model adapted via ISO can mine geometric knowledge about the target distribution and thus generalize well to it, without requiring any prior for regularization [12] or post-processing [57].

**Representation learning.** To further analyze how ISO helps during inference, we train a 2-way classifier to predict which dataset (Human3.6M or 3DHP) a given 3D pose comes from. The classifier after trained can achieve averagely 99.5% accuracy on both datasets, demonstrating the classifier's ability to accurately capture the inter-dataset difference of geometry and judge the dataset (or distribution) a 3D pose comes from. Then, we apply this classifier to distinguish whether the 3D poses estimated by *Baseline* and ISO are close to the distribution of GT 3D poses from 3DHP. The classifier only identifies 52.6% of the 3D poses estimated by *Baseline* drawn from the target 3DHP distribution, while 83.4% of the 3D poses estimated by ISO drawn from 3DHP. This demonstrates the representations adapted by ISO are more similar to the target ones. Additionally, we visualize the hidden feature (1024-dimension vector) of the classifier by t-SNE [29] in Fig. 5. We can see performing SSL on target instances draws the feature distribution of the generated 3D poses closer to those of GTs (blue and green circles). All these results clearly demonstrate ISO enables the model to adapt to the real distribution of 3DHP during inference stage.

**Per body-part improvement.** In addition to distribution alignment, we also study the performance improvement of our method on each body part. We first divide all skeleton joints into eight parts: Hip, Spine, Shoulder, Head, Elbow, Wrist, Knee and Ankle. Then we compute mean 3D PCK for each part and present the results in Fig. 6. We can see *Online* improves over *Baseline* by a large margin for Head, Elbow, Wrist and Ankle. All these parts are difficult to estimate especially for samples from new scenarios, due to high flexibility. However, *Online* successfully estimates these parts, which demonstrates the effectiveness of our method for cross-scenario generalization.

## 4.4 Is ISO costly or sensitive to noise?

**Inference time analysis.** Our ISO scheme is slightly slower than a regular inference scheme, which only performs a single forward pass for each sample. Here, we provide two potential solutions to

Table 3: Inference time analysis of different inference modes of ISO.

| Method | PCK | AUC | MPJPE | Time[s] |
|--------|-----|-----|-------|---------|
| Vanilla | 82.5 | 47.6 | 94.1 | 0.244 |
| Vanilla-lr | 82.1 | 47.3 | 94.6 | 0.027 |
| Online | 83.6 | 48.2 | 92.2 | 0.027 |
| Online-skip | 83.0 | 48.0 | 92.7 | 0.004 |
| Baseline | 78.9 | 43.7 | 103.8 | 0.003 |

Table 4: Performances with different levels 2D pose noise from $\mathcal{N}(0, \sigma)$.

| Method | PCK | AUC | MPJPE |
|--------|-----|-----|-------|
| ISO | 83.6 | 48.2 | 92.2 |
| ISO ($\sigma = 5$) | 82.5 | 47.4 | 94.0 |
| ISO ($\sigma = 10$) | 79.6 | 43.7 | 103.4 |
| Baseline | 78.9 | 43.7 | 103.8 |

improve the computational efficiency. For vanilla ISO, we set iteration $T$ to 1 (instead of 10) and learning rate $\alpha$ to $2e^{-4}$ (instead of $2e^{-5}$). The new setup is denoted as *Vanilla-lr*. For online ISO, since $T$ is already 1, we propose to perform SSL once per 10 samples, denoted as *Online-skip*. For all settings, we count average per-sample inference time in seconds and show results in Table 3.[2] We observe by adopting the new inference setup, the computational efficiency can be improved by nearly $8\times$ and $7\times$ speedup for vanilla and online ISO, respectively, with good performance almost the same as the original. Significantly, we see *Online-skip* achieves almost the same efficiency as the regular inference scheme while improving the performance by a large margin.

**Robustness to noisy observations.**    We evaluate robustness of our method under different levels of noise by adding noise to the input 2D poses. Specifically, we add Gaussian noise $\mathcal{N}(0, \sigma)$ to the GT 2D poses, where $\sigma$ is the standard deviation in pixel [30]. The results are shown in Table 4. The accuracy decreases linearly with $\sigma$, which indicates the noise of 2D poses has major impact on the results. However, this issue can be alleviated by using state-of-the-art 2D pose estimators [35, 43, 10] or training with synthetic error [33, 6]. Note the maximum person size from head to foot is approximately 200px in the input data. Thus, Gaussian noise with $\sigma = 10$ is considered as extremely large. However, even under such large noise, ISO produces a better result (79.6% 3D PCK) than *Baseline* (78.9% 3D PCK), which verifies its robustness.

# 5   Conclusion

We propose a new ISO framework for improving the generalizability of 3D pose estimation models. It explores underlying priors in target instances and leverages SSL techniques to mine such knowledge for estimating 3D poses accurately even under strong distribution shifts between source and target scenarios. ISO achieves state-of-the-art performance on challenging MPI-INF-3DHP benchmark under cross-scenario setting. In future, we plan to investigate more SSL techniques in our framework.

# Broader impact

We propose Inference Stage Optimization (ISO) framework for cross-scenario 3D human pose estimation. It can be applied to lots of 3D pose estimation related applications including human-robot interaction, action recognition, human tracking, etc., which are all important research topics in artificial intelligence. However, similar to most human pose estimation methods, ISO may be used for military application and raise privacy concerns when misused. Generally, improving generalization performance for the 3D human pose estimation task may have many applications, which could be positive, negative or more complicated, but would depend on what task we use these applications for.

# Acknowledgements

Jiashi Feng was partially supported by MOE2017-T2-2-151, NUS_ECRA_FY17_P08, AISG-100E-2019-035 and CRP20-2017-0006. The authors would like to thank Chenyang Si for helpful discussions and comments.

## Footnotes

[1] Implementation is based on source code: SemGCN and RepNet for [55] and [51], respectively.

[2]The time is counted on single GPU TITAN X and CPU Intel I7-5820K 3.3GHz.

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
