[Supplementary Material]

# Inference Stage Optimization for Cross-scenario 3D Human Pose Estimation
# (Supplementary Material)

**Jianfeng Zhang   Xuecheng Nie   Jiashi Feng**

Department of ECE, National University of Singapore

zhangjianfeng@u.nus.edu niexuecheng@u.nus.edu elefjia@nus.edu.sg

## 1  Ratio of limb length calculation

We compute the limb length ratios of upper to lower arm and leg (both for the left and right sides) as well as torso, for geometric distribution analysis. The joints and body parts of interest are defined in Fig. S1. The ratio of arm is computed as $\frac{\text{limb(elbow} \rightarrow \text{shoulder})}{\text{limb(wrist} \rightarrow \text{elbow})}$, both for the left and right body sides, where $\text{limb}(c \rightarrow p)$ represents limb length between child joint $c$ and parent joint $p$ of the skeleton in Fig. S1. Similarly, the ratio of leg is computed as $\frac{\text{limb(knee} \rightarrow \text{hip})}{\text{limb(ankle} \rightarrow \text{knee})}$, both for the left and right sides. The ratio of torso is computed as $\frac{\text{limb(thorax} \rightarrow \text{spine})}{\text{limb(spine} \rightarrow \text{pelvis})}$.

Figure S1: A human pose represented as skeleton joints and bones. [4]

## 2  Component analysis

In this section, we analyze the impact of SSL techniques and hyper-parameters on cross-scenario 3D pose estimation accuracy. All the results are reported under *unscaled* protocol.

**How does the choice of self-supervised learning technique impact accuracy?**  We first study the influence of different SSL techniques on the model's generalizability. We use *Adv* and *Cyc* to represent **ISO-Adversary** and **ISO-Cycle** SSL techniques, respectively. The results are shown in Table S1. We can observe *Adv* (*Joint*, *Vanilla* and *Online* settings) improves accuracy upon *Baseline* by a large margin. In addition, we observe *Cyc* achieves even better results than *Adv* on all three settings by adding additional geometric cycle consistency constraint. These observations demonstrate the importance of adversarial learning and geometric knowledge to cross-scenario 3D pose estimation, which may motivate more SSL techniques in the future.

Table S1: Ablation of different SSL techniques on MPI-INF-3DHP.

| Method | PCK | AUC | MPJPE |
|---|---|---|---|
| Baseline | 78.9 | 43.7 | 103.8 |
| Joint-Adv | 80.9 | 46.1 | 97.0 |
| Vanilla-Adv | 82.1 | 47.2 | 95.3 |
| Online-Adv | 83.0 | 47.6 | 93.1 |
| Joint-Cyc | 81.3 | 46.9 | 96.2 |
| Vanilla-Cyc | 82.5 | 47.6 | 94.1 |
| Online-Cyc | 83.6 | 48.2 | 92.2 |

**How does hyper-parameters impact accuracy?**  We then analyze the sensitivity of our method to hyper-parameters i.e., learning rate $\alpha$ and training iteration $T$ used when performing ISO (*Cyc*). Specifically, we report 3D PCK for both vanilla and online ISO, and show the results in Fig. S2. We

Figure S2: Analysis on hyper-parameters. Left: Training iteration $T$. Right: Learning rate $\alpha$. For online ISO, the best $T$ and $\alpha$ are set to 1 and $2e^{-5}$. Further increasing them causes poor performance. For vanilla ISO, the best $T$ and $\alpha$ are set to 10 and $2e^{-5}$.

first analyze the impact of $T$ by varying $T$ while fixing $\alpha$ to $2e^{-5}$. From Fig. S2 (Left) we can observe that increasing $T$ from 1 to 10 for vanilla ISO, the accuracy is consistently increased from 81.5% PCK to 82.5% PCK, due to the geometric knowledge mined from the target instances. However, further increasing $T$ degrades the performance, caused by overfitting to the SSL task. We can also see that the model adapted under online ISO achieves best performance 83.6% PCK when $T = 1$, and the performance decreases when adopting a larger $T$. The main reason is performing SSL under online mode with $T > 1$ will make the model quickly overfit to the SSL task, thus hamper 3D pose estimation. Then we fix $T$ to 10 and 1 for vanilla and online ISO, respectively, and apply different $\alpha$ (ranging from $2e^{-3}$ to $2e^{-7}$) to study the influence of learning rate on performance. Fig. S2 (Right) shows that both modes achieve best performance when $\alpha = 2e^{-5}$. Further decreasing learning rate, the performance of both modes gradually degrades and gets close to *Joint* (i.e., the model without adapting) with 81.3% PCK. However, performing ISO with a large $\alpha$ (e.g., $2e^{-3}$), the accuracy quickly drops, especially for online mode (70.9% PCK), since training with a large learning rate, the model easily overfits to the SSL task and thus restricts performance.

## 3 More qualitative results

Here more visual results obtained by ISO on LSP [2], MPII [1], MPI-INF-3DHP [3], and 3DPW [5] are shown in Fig S3. From Fig S3 we can observe that ISO can produce satisfactory 3D pose estimations even in case of large occlusion (first and second columns) and strong pose deformations (third and fourth columns). All these results clearly demonstrate the excellent generalizability of ISO to new target scenarios.

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

Figure S3: Examples of ISO results for 3D pose estimation on the challenging cases from the LSP, MPII (first three rows), and 3DHP, 3DPW (last row).