[Reviews · NeurIPS 2020]

Review 1

Summary and Contributions: This paper presents an interface stage optimization framework to adapt 3D human pose estimation to novel scenarios with enhanced generalization. With both fully supervised learning and self-supervised learning co-trained on source data, the proposed ISO performs self-supervised adaption to each inference instance to accommodate distribution shifts between cross scenarios. It has achieved improved accuracy on cross-scenario setup.

Strengths: The paper has a novel framework for practical cross-scenario 3D human pose estimation and extensive experiments have been done demonstrating the effectiveness of the method. The framework is orthogonal to different self-supervised learning techniques and could inspire more future works on domain adaption for human pose estimation.

Weaknesses: My major concern towards this method is the inference time performance. Even though the paper has proposed a few speedup solutions, the vanilla-lr is still 9x slower than the baseline (and the numbers in other tables should be based on the faster version for a fair comparison). I am wondering how the accuracy would be affected if the parameters adaption could be performed on a batch of interference images (several images altogether or even a dataset) instead of on each image instance. How necessary is this *individual* adaption?

Correctness: Seems good.

Clarity: yes, I find it easy to follow.

Relation to Prior Work: Good. I find this recent paper also related: "Weakly Supervised 3D Human Pose and Shape Reconstruction with Normalizing Flows"

Reproducibility: Yes

Additional Feedback: - It is not very clear to me how the pose discriminator are trained and updated during inference? What's the training data at both stages? Could it be unbiasedly "adapted" based on a single image instance during inference? - What's the inference accuracy with and without ISO over Human3.6M data? It would be good to have such an experiment to understand if ISO would help even when there is no pose distribution shift.


Review 2

Summary and Contributions: This paper proposed an Inference stage optimization (ISO) method for improving the generalizability of 3D pose estimation model. The proposed method performs geometry-aware self-supervised learning to update the neural network model for each instance. Two SSL techniques, i.e., ISO-Adversary and ISO-Cycle, are adopted to the 3D pose estimation task under the IOS framework. The experiments proved that the proposed method can improve the generalizability of the 3D pose estimation model.

Strengths: This paper focused on an interesting problem of improving 3D pose model generalizability in the inference stage. It adopted two SSL methods, ISO-Adversary and ISO-Cycle ,to update model during inference. The experimental results show the effectiveness of the proposed method.

Weaknesses: The inference stage optimization seems similar to the previous model-fitting methods, e.g., Kolotouros et al. Learning to reconstruct 3D human pose and shape via model-fitting in the loop. CVPR 2019. However, the comparisons are not presented. Ablation studies in the supplementary material show that the main improvements are from the joint training. In MPJPE metric, the joint-adv improves the baseline by -6.8 and the inference stage optimization vanilla-adv improves the model by -1.7 (vs joint-adv). As supervised learning and weakly supervised learning have been studied in the previous papers[7, 9, 35], the comparison should be presented in table 1 and table 2.

Correctness: The proposed method is technically correct.

Clarity: The paper is well written and easy to follow. It is suggested that the implementation details and ablation study parts should be moved from the supplementary material to the main paper.

Relation to Prior Work: The difference between the proposed method and the prior works is not clearly discussed. The relation to the unsupervised or weakly supervised methods, and model-fitting based methods should be highlighted.

Reproducibility: Yes

Additional Feedback: The analysis in section 4.3 seems interesting. It is better to visualize the ground truth in figure 4. In the robustness analysis, some images should be presented to show how the noise influences the input and the output.


Review 3

Summary and Contributions: The authors propose a neural network training scheme that finetunes a pre-trained 3D pose estimation model on unlabelled target images (a form of transductive learning) in an iterative fashion.

Strengths: Focussing on geometry instead of appearance distribution shift between domains makes sense, as good 2D detectors exist. I really liked the detailed analysis of why the proposed methods help and what their effect is, .e.g., on capturing the bone length ratios in the target distribution. The online training (alongside prediction) is a neat idea and effective in speeding up the adaption to the target distribution. The cycle consistency loss and re-projection adversarial losses have been used elsewhere; yet, their application to the transductive case and making this effective and efficient is nontrivial. The gained improvement is impressive across the board.

Weaknesses: It is not explained what 2D pose is used as input. Is it the GT pose or one estimated by OpenPose or similar off-the-shelf method? This is the most important concern I have, please clarify in detail. Other self-supervised methods are not evaluated in the transductive setting (seeing examples of the target dataset), would this be possible with the available codebases? It would be good to mark in the table which ones are transductive (e.g. [A] should be added as it is transductive too).

Correctness: The method is mostly a system of existing components that are tied together in a neat way. The individual components and the overall approach is sound.

Clarity: The paper is clearly written and easy to understand.

Relation to Prior Work: Methods for transductive learning should be discussed. Particularly the dollowing one doing it for human pose estimation. The intro needs to be changed in that this method is not the first to do transductive learning for human pose estimation. [A] Neural Scene Decomposition for Multi-Person Motion Capture Rhodin et al. CVPR 2019

Reproducibility: Yes

Additional Feedback: It is great that additional results and code was given in the supplemental. Potential negative effects when misused should be discussed in the impact section. Update: Thanks for the rebuttal, I stand to my score but it is a bummer that all evaluation is on ground truth. If possible, include an experiment on 2D pose given from OpenPose or similar.


Review 4

Summary and Contributions: The paper proposes Inference Stage Optimization(ISO) to incorporate self-supervised learning(SSL) for lifting 2D body pose to 3D body pose into the lifting network trained with full supervision using 3D pose label to help the generalization of the network to target data having different pose distribution. During the inference stage, SSL is also optimized for the new input before making a prediction for the 3D body pose. The experiments show substantial improvements in the metrics compared with existing work and the baseline.

Strengths: Applying supervised learning based method in the inference stage to transfer the trained model for data from different distributions. Substantial improvements over the baseline and existing methods in making prediction for data from different distributions. Analyses on why ISO works provide some intuitions about the efficacy of the proposed methods.

Weaknesses: Though the authors shadow many insights on why ISO performs well, I still have questions about the Shared Feature Extractor, SSL Head, FSL Head. As the SSL is from existing work and the main contribution is combination of SSL with FSL, answering the questions clearly is important. Which kind of feature, information is shared in the Shared Feature Extractor? How much will it divert when trained on new target data so that is causes the FLS head fail? What information is kept in the FSL head? How much source data specific information is in the FSL head? What information is kept in the SSL head? Do SSL and FSL heads share some common information? Or the features are well disentangled? The other issue is the online prediction for the new coming input. As the data comes in sequentially, how to learn the network for new instances without overfitting to those come first becomes an issue and also an important question to answer. Though the authors have a vanilla ISO version to adapt the Shared Feature Extractor on each coming instance which has already show improvements over the baseline, it's not adequate to answer this question. In the supplementary material, Fig S2 shows the online ISO is very sensitive to the learning rate and training iteration which agrees with my thinking. Line55 to Line 58 attribute this to "the model quickly overfit to the SSL task, thus hamper 3D pose estimation". The explanation is ambiguous and I don't understand the SSL task and 3D pose estimation mean. In essence, SSL also makes 3D pose estimation and SSL and FSL should agree with each other. My answer for this sensitiveness is the network overfit to the sequentially coming data and can coverage to the later data, which is supported by the fact that Vanilla ISO is not sensitive to the training iteration and learning rate.(Fig S2) Minors: A brief introduction or grouping of the methods in Table 1 would give better idea of the improvements by the contributions.

Correctness: Yes

Clarity: Yes

Relation to Prior Work: Yes.

Reproducibility: Yes

Additional Feedback: ===UPDATE AFTER REBUTTAL'=== Thank the authors for the rebuttal. The rebuttal has explained most of the questions. I raise the original rating above the acceptance threshold.

[Author Response · NeurIPS 2020]

| Table R1 | | | | |
|---|---|---|---|---|
| Method | PCK | AUC | MPJPE | Time[s] |
| Vanilla-S | 82.5 | 47.6 | 94.1 | 0.244 |
| Vanilla-B | 83.7 | 48.4 | 91.3 | 0.083 |
| Online-S | 83.6 | 48.2 | 92.2 | 0.027 |
| Online-B | 84.1 | 48.7 | 90.5 | 0.007 |

| Table R2 | |
|---|---|
| Method | MPJPE |
| Baseline | 45.0 |
| Joint | 42.6 |
| ISO | 41.8 |

| Table R3 | | | |
|---|---|---|---|
| Method | PCK | AUC | MPJPE |
| [7] | 65.3 | 33.4 | 135.7 |
| [9] | 75.9 | 36.3 | - |
| [35] | 62.4 | 30.9 | 147.4 |
| Ours | 83.6 | 48.2 | 92.2 |

| Table R4 | | | |
|---|---|---|---|
| Method | PCK | AUC | MPJPE |
| [14] | 60.9 | 30.2 | 150.1 |
| [7] | 65.3 | 33.4 | 135.7 |
| Ours | 83.6 | 48.2 | 92.2 |

**To R1  Q1. Batch adaptation.** ISO can perform model adaptation with a batch of instances, if available. See in
Table R1, where **B** and **S** denote batch and single-instance ISO. Batch ISO outperforms single-instance ISO in both
accuracy (from 94.1 to 91.3 in MPJPE for *Vanilla*) and efficiency ($\sim 3\times$ and $4\times$ faster for *Vanilla* and *Online*).
**Q2. Comparison based on faster version of ISO** will be added in the revision.
**Q3. The pose discriminator** is pretrained on the source data. During inference, it is updated using Eqn. (3) for each
new target instance. To alleviate potential training bias from a single instance, we applied horizontal flip augmentation
and the augmentation strategy (see Line 159-161) to update the discriminator. We will add more details in the revision.
**Q4. Ablation studies on w/ and w/o ISO over H36M** are given in Table R2. We observe *Joint* improves *Baseline* by
a large margin. ISO makes marginal improvement over *Joint* (from 42.6 to 41.8 in MPJPE) since training and testing
distribution are similar. However, even though there is no significant distribution shift, ISO still makes positive effect.

**To R2  Q1. Differences between ISO and model-fititng (MF) methods**:  1) The motivation and methodology are
different.  The MF methods (e.g., Kolotouros et al., CVPR 2019) aim to improve the model training by iterating
regression (using a parametric human body model) and optimization. However, they do not consider how to generalize
the model to new testing distributions that are different from the training ones. Our ISO focuses on mining distributional
knowledge about the testing distribution from unlabeled instances via SSL and adapt the models accordingly to gain
better generalizability. 2) The MF methods usually require a parametric model (e.g., SMPL) and a 3D body mesh
initialization for model fitting; whereas ISO does not require these and thus is more general.
**Q2. "Main improvements come from *Joint*"** is only true for *Vanilla* as *Vanilla* is always re-initialized using *Joint* for
each new instance. Thus, *Vanilla* cannot benefit the model too much as distributional information mined from a single
instance is limited. When more distributional information is mined, ISO can make significant improvement. This can
be observed from: 1) *Online* improves *Joint* by -4 in MPJPE (see Table S1). 2) Batch version of *Vanilla* improves the
performance upon single-instance version of *Vanilla* by a large margin (see **Q1** in responses to **R1** and Table R1).
**Q3.Differences from un-/weakly-supervised methods:** un-/weakly-supervised methods are usually used for better
training the model with more unlabeled *training data* (see Line 72-78) and rarely used in a transductive manner for
testing; while ISO aims to improve the model's generalizability during inference via SSL. Among the suggested methods
[7, 9, 35], [7] uses cycle-supervision and can be used in a transductive manner; while [9,35] requires multi-view data
which are not available on 3DHP and 3DPW during testing. We compare ISO with them on 3DHP in Table R3. ISO
significantly outperforms them. Comparison with them on 3DPW will be added in revision.
**Q4. Qualitative results.** We will visualize GT in figure 4 and show some results with noisy inputs in the revision.

**To R3  Q1. 2D pose input.** For fair comparison with [11, 46, 50], in Table 1, we use GT poses as inputs. However,
ISO is robust to 2D pose noise as shown in Table 4 and thus can perform similarly well when using detected poses.
**Q2. SSL methods under transductive learning (TL) setup.** [A] cannot be evaluated under TL setup on 3DHP since
it requires multi-view data while the test set only has single-view data. We re-implement other SSL methods [7, 14] and
evaluate them on 3DHP under TL setup (see in Table R4). We can observe ISO outperforms them by a large margin.
**Q3. Relation between ISO and TL methods** will be added in the revision.

**To R4  Q1. Questions about model components.** To study these, we visualize
the hidden features of different samples (random samples from H36M test set and
samples from 3DHP) extracted by both heads using t-SNE (see right figure). The
samples from 3DHP are the ones showing improvement after ISO, which present
novel viewpoints and body sizes.  We observe features from SSL head of both
datasets (green & blue squares) are closer than the ones from FSL head (orange
& red circles). Thus, SSL head learns features more robust to distribution change
whereas FSL head learns more discriminative features from source data.  The
features from these two heads are thus disentangled. Shared feature extractor keeps
these two kinds of features. FSL head would fail when facing unusual poses and
viewpoints (e.g., top views) which are very ambiguous with only monocular 2D pose input.

**Q2. Will online ISO overfit to the samples come at first?** To answer this, we randomly shuffle the test set of 3DHP
before performing online ISO. We conduct experiments for 8 times and obtain the statistics: PCK: $83.2 \pm 0.43$, AUC:
$48.0 \pm 0.30$ and MPJPE: $92.9 \pm 1.3$. The small variance implies online ISO does not overfit to the sequential data. As
discussed in **Q1**, the model overfitting to the SSL task would extract less discriminative features and thus hamper the
performance. We will clarify this in the revision.

[Meta-Review · NeurIPS 2020]

This paper proposes and inference stage optimization to improve 3d human pose estimation. All reviewers recommend acceptance but the paper can benefit from additional analysis and clarity with respect to existing work. Please include the additional references and provide the clarifications requested during reviewing.